# Endoscopic Imaging Technology Today

**DOI:** 10.3390/diagnostics12051262

**Published:** 2022-05-18

**Authors:** Axel Boese, Cora Wex, Roland Croner, Uwe Bernd Liehr, Johann Jakob Wendler, Jochen Weigt, Thorsten Walles, Ulrich Vorwerk, Christoph Hubertus Lohmann, Michael Friebe, Alfredo Illanes

**Affiliations:** 1INKA Health Tech Innovation Lab., Medical Faculty, Otto-von-Guericke University Magdeburg, 39120 Magdeburg, Germany; michael.friebe@ovgu.de (M.F.); alfredo.illanes@med.ovgu.de (A.I.); 2Clinic of General-, Visceral-, Vascular- and Transplant Surgery, University Hospital Magdeburg, 39120 Magdeburg, Germany; cora.wex@med.ovgu.de (C.W.); roland.croner@med.ovgu.de (R.C.); 3Uro-Oncology, Roboter-Assisted and Focal Therapy, Clinic for Urology, University Hospital Magdeburg, 39120 Magdeburg, Germany; uwe-bernd.liehr@med.ovgu.de (U.B.L.); johann.wendler@med.ovgu.de (J.J.W.); 4Hepatology, and Infectious Diseases, Clinic of Gastroenterology, University Hospital Magdeburg, 39120 Magdeburg, Germany; jochen.weigt@med.ovgu.de; 5Clinic of Cardiac and Thoracic Surgery, University Hospital Magdeburg, 39120 Magdeburg, Germany; thorsten.walles@med.ovgu.de; 6Clinic of Throat, Nose, and Ear, Head and Neck Surgery, University Hospital Magdeburg, 39120 Magdeburg, Germany; ulrich.vorwerk@med.ovgu.de; 7Department of Orthopedics, University Hospital Magdeburg, 39120 Magdeburg, Germany; christoph.lohmann@med.ovgu.de; 8Department of Measurement and Electronics, AGH University of Science and Technology, 31-503 Kraków, Poland

**Keywords:** endoscopy, technology, trends, medical technology, endoscope provider, future technologies, artificial intelligence, big data, image analysis, image processing

## Abstract

One of the most applied imaging methods in medicine is endoscopy. A highly specialized image modality has been developed since the first modern endoscope, the “Lichtleiter” of Bozzini was introduced in the early 19th century. Multiple medical disciplines use endoscopy for diagnostics or to visualize and support therapeutic procedures. Therefore, the shapes, functionalities, handling concepts, and the integrated and surrounding technology of endoscopic systems were adapted to meet these dedicated medical application requirements. This survey gives an overview of modern endoscopic technology’s state of the art. Therefore, the portfolio of several manufacturers with commercially available products on the market was screened and summarized. Additionally, some trends for upcoming developments were collected.

## 1. Introduction

Endoscopy is the optical inspection of the inner structures of something. In medicine, it describes the procedure of inspection of the inner cavities of the human body. It is a technology with a long tradition beginning already in the antique with rectal speculums. The story of modern endoscopy started in the early 19th century when the light came into the darkness. Bozzini introduced a mirror to reflect the light of a candle directly into the direction of view through the hollow endoscope tube into the body [1]. After that, several groundbreaking inventions turned endoscopy into the most used imaging modality in medicine today. This covers the introduction of mirrors and electric light, endoscopes with lens systems, the first semi-flexible and flexible endoscopes, the combination with camera systems, the integration of cameras into the endoscope, the combination with minimally invasive treatment options, the use of enhancement technologies and, nowadays the introduction of Artificial Intelligence for decision support, to name only a few.

Additionally, the basic principle of optical inspection was adapted to a broad field of medical applications, resulting in many distinct designs and functionalities. Modern endoscopy’s state of the art is developing fast, benefiting from advances in electronics, computing, and digitalization. This survey presents the state of the art of medical endoscopy as it is used today in clinical routine in various fields of application. Therefore, the catalogs of the most visible 20 providers of endoscopic imaging systems were screened, medical users were interviewed, and some trends for upcoming technologies were collected. This paper aims to give an overview of the field of applications, the general design and functionalities of endoscopic imaging systems, the technical features and cutting-edge technologies provided by manufacturers.

## 2. Research Strategy for Data Collection

The clinical application fields of endoscopic imaging and the 20 most visible providers of endoscopic imaging systems were researched on the internet covering general websites, articles [2,3] or market reports [4,5]. This led to an overview of the medical applications and allowed screening of the product portfolio of the endoscopic imaging providers [6,7,8,9,10,11,12,13,14,15,16,17,18,19,20,21,22,23,24,25]. The results were collected, sorted, analyzed and documented in Section 2.1, Section 2.2, Section 2.3 and Section 2.4 of this paper. A deeper analysis of the product portfolio and interviews with medical professionals of various medical disciplines were used to summarize endoscopic technology’s state of the art in daily routine. Again, for this paper’s “upcoming trends” section, the providers’ catalogs were analyzed for cutting-edge technology and advertised changes in their portfolios. A PubMed research was utilized to select papers showing new technology with a close-to-market state. The presented endoscopic technology sketches the status of endoscopy at the beginning of the year 2022. It does not claim to be all-embracing or complete.

Keywords used for Internet and PubMed research: Endoscop * + healthcare, medical, technology, imaging, design, processing, Artificial Intelligence, Big Data, image, atlas, repository, database, first, study.

Publication Date: last two years, January 2019–January 2022.

### 2.1. Endoscopy—A Short Introduction

Endoscopy is a medical procedure to visualize the human body’s internal organs or natural cavities [26]. The endoscope is a rigid or flexible tubular device allowing a direct view into the body. An endoscopic system can be built up as a purely optical device containing lenses, transparent rods or fibers or in combination with integrated or add-on cameras. The endoscope tip can be inserted through a small access created by incision or through the natural lumina of the body. A light source provides sufficient illumination to the examined cavity. When cameras are used, the field of view can be visualized on a screen and recorded for later diagnosis or documentation. There are several different types of endoscopic systems to examine, e.g., the oral cavity, joints, lung, abdomen, bladder or colon. Besides diagnostics, endoscopes are used to support therapeutic procedures like surgery or minimally-invasive interventions.

### 2.2. Common Endoscopic Procedures

Medical endoscopes are usually named related to the field of their application, organ or procedure [27]. Known applications are laparoscopy, thoracoscopy, gastrointestinal endoscopy including esophagoscopy, gastroscopy, duodenoscopy, colonoscopy and small bowel endoscopy, bronchoscopy, arthroscopy, urological endoscopy, gynecological endoscopy, ear, nose and throat endoscopy or neuro- endoscopy.

#### 2.2.1. Laparoscopy

In a laparoscopy, small incisions are made in the abdominal wall, introducing a scope into the peritoneal cavity through a trocar. It can be done for diagnosis and as image guidance for surgical procedures [28]. For surgical procedures, typically, rigid endoscopic optics are used. They can be handled manually or connected to an assistance system or robotic manipulators.

#### 2.2.2. Gastrointestinal Endoscopy

Gastrointestinal endoscopy procedures are separated into esophagoscopy, gastroscopy, duodenoscopy, colonoscopy and small bowel endoscopy [29].

##### Esophagoscopy, Gastroscopy, Duodenoscopy (EGD)

Esophagoscopy, Gastroscopy and Duodenoscopy are a subgroup of gastrointestinal endoscopy. These procedures are performed to diagnose and treat diseases of the upper digestive system. The flexible tubular gastroscope or duodenoscope is fed through the mouth to visualize the esophagus, the stomach and the duodenum. Gastroscopes and duodenoscopes can have an additional working channel to introduce accessory devices.

Endoscopic retrograde cholangiopancreatography (ERCP) is a hybrid technique of endoscopy and radiological intervention and also allows cholangioscopy with dedicated endoscopes through the working channel of the duodenoscope.

##### Colonoscopy

A colonoscopy is a procedure to examine the colon. The colonoscope is an extended, flexible, tubular scope. It is inserted through the rectum and can be advanced up to the end of the large intestine and even partially into the adjacent small intestine. Besides imaging, polyps or other lesions can be removed during the same examination by performing endoscopic resections that are not limited by the diameter of the working channel.

##### Small Bowel Endoscopy

There are different technologies for imaging the small bowel. For capsule endoscopy, a pill-sized capsule containing batteries, a light source and one or more cameras is used. The capsule can be swallowed and travels through the digestive tract. Between 50- and 100-thousand images are captured in a frame rate from 1–30 Hz. The images are transmitted wireless to a device outside the body or stored in a flash memory of the capsule. After 24–48 h, the capsule leaves the body through the rectum.

Other methods to image the small bowel are single-balloon-, double-balloon- and spiral enteroscopy. Special over tubes with inflatable balloons or a motorized spiral are used to pull and pleat the intestine onto the enteroscope.

#### 2.2.3. Bronchoscopy

The thin, flexible bronchoscope is fed through the mouth or nose into the trachea and the bronchial tubes in a bronchoscopy. The procedure serves to diagnose tumors or other lung diseases, collect biopsies under image guidance or manage airway narrowing or bleedings [30].

#### 2.2.4. Thoracoscopy

Thoracoscopy is the endoscopic examination of the pleural cavity or the lungs through a small incision in the chest wall. Rigid or semi-flexible endoscopes with diameters o 2–10 mm are introduced through a trocar to inspect the pleural space for biopsy or diagnosis and management of pleural effusion. The thoracoscope can have a working channel for drainage or to use biopsy forceps and other devices.

#### 2.2.5. Mediastinoscopy

A mediastinoscopy is a procedure to diagnose the mediastinum, a space in the center of the chest that contains the heart, the main blood vessels, lymph nodes and the esophagus. The mediastinoscope has a blade that is entered into this space and that opens up the access path to the situs. A lens optic and light channel or a chip on the tip camera and light source are integrated into this blade. Some blades can be spread to provide more space to ease biopsy or lymph node extirpation.

#### 2.2.6. Arthroscopy

An arthroscopy examines joints through a tiny incision in the skin for diagnostics or to observe minimally invasive treatments in joints. Most commonly, arthroscopy is performed in the knee, shoulder, elbow, wrist, ankle, foot, or hip joint due to the accessibility and diameter of the rigid arthroscope [31]. For smaller joints, nowadays, small flexible arthroscopes are available.

#### 2.2.7. Urological Endoscopy

Urological endoscopy is divided into urethrocystoscopy to inspect the urethra and bladder (lower urinary tract) and ureterorenoscopy to image the ureter or renal pelvis and calyces (upper urinary tract) [32]. Both can be performed with rigid or semirigid and flexible systems. Ureteroscopes are thinner and flexible to enter the ureter from the bladder. The pelvicalyceal system can also be imaged by percutaneous nephroscopy access.

#### 2.2.8. Gynecological Endoscopy

Hysteroscopy is a non-surgical way to diagnose or treat the inner uterus via vaginal access [33]. A hysteroscope can be a rigid or flexible instrument. Flexible hysteroscopes provide smaller diameters and allow examination without prior cervical dilation.

#### 2.2.9. Ear, Nose and Throat Endoscopy

Ear, Nose and Throat (ENT) endoscopy focuses on examining the oral cavity, nasal passages, oropharynx down to the larynx, openings to the sinuses, and the middle ear. Rigid or flexible scopes are in use depending on the access and insertion depth. Rigid scopes allow the view through straight pathways. In contrast, flexible scopes can be fed through the nose and even down to the larynx in the same procedure covering the areas of the oropharynx, nasopharynx, larynx down to the trachea and esophagus.

#### 2.2.10. Neuro-Endoscopy

Neurosurgeons perform Neuro-endoscopy in minimally invasive procedures and surgical approaches to image the intraventricular space. A trans-nasal neuro-endoscopic approach can be used for micro-neurosurgery, e.g., on the pituitary gland.

#### 2.2.11. Robotic Endoscopy Systems

Robotic endoscopy systems are connected to manipulation systems for robotic-assisted surgery. These manipulation systems allow control of the field of view or insertion depth via drives and a steering console [34]. Depending on the robotic system, the type of surgery and the access path to the structure of interest, the endoscopes can be rigid or flexible. Robotic-driven systems enable the control of multiple tools and the visualization with more degrees of freedom by a single user.

### 2.3. Endoscopy Systems

Today the standard of endoscopy in diagnostics and therapy is digital imaging. Endoscopes are connected to digital imaging systems allowing visualization on screens and photo or video documentation. Even if the design and type are highly dependent on the intended use, a general structure of components can be summarized. The main component of an endoscopy system is the endoscope itself which is the part that can be introduced into the human body. The endoscope can be flexible or rigid and covers dimensions from the submillimeter range up to 25 mm. A camera can be connected to the endoscope proximal end by standardized connectors [35] or is integrated into the endoscope. Endoscopes with camera sensors placed on the tip of the endoscope are usually called “chip on the tip” or “video” endoscopes. Endoscopes containing rod lenses or fibers are called “Hopkins” endoscopes. The endoscopic camera is connected to a video processing unit to control the image acquisition. The connectors, therefore, are not standardized and differ depending on the provider of the system and even the imaging system generation. The basic functionalities of a video processing unit are “start” and “stop” of the image acquisition, storage, change of visualization parameters, processing, annotations and presentation of the image on a screen. The third relevant part of an endoscopic imaging system is the light source. Besides the power to sufficiently illuminate the inspected cavity, the homogenous distribution, color and an option for adoption to avoid overexposure are essential parameters for light sources. Advanced light sources offer separation or filtering of a different wavelength to improve the enhancement of structures of interest either via filtering light or composing light on the light-emitting side or on the detection site, which is then called postprocessing.

Endoscopy systems can be combined with several accessories such as holders, suction, sterile covers, introducer sheets, trocars, cleaning, or therapeutic instruments like graspers or scissors.

The essential components of an endoscopic system and an example of a workspace for ENT endoscopic diagnostics are shown in Figure 1.

### 2.4. Endoscope Providers and Technologies

There are several endoscopes and endoscopy systems providers on the market. Those can be distinguished between whole system providers, endoscope providers, accessories, and consumable providers. Most companies on the market limit their offer to a specific range of applications or medical fields. In Table 1, the 20 most visible endoscope and endoscopy system providers are listed in alphabetic order. The fields of application and portfolio are added based on information on the provider’s website or catalog. Additionally, the provider’s key technologies or technology highlights are summarized. The table illustrates an overview but does not claim to be complete.

### 2.5. State of the Art Technology 2022

#### 2.5.1. Endoscopes

Rigid endoscopes are used for applications with straight access paths. The diameter and length depend on the intended use of the endoscopes. For example, rigid endoscopes for laparoscopic applications can have a diameter of 5, 10 or 12 mm and a length between 300 and 500 mm. For neuro or nose applications, diameters are between 2 and 4 mm with a short length of 60–200 mm. Some variants of rigid endoscopes are shown in Figure 2a. Common designs show the following components (Figure 2b): The main body is usually made out of massive stainless steel, the eyepiece made out of plastic, a stainless steel pipe as the shaft and housing for the optical path and a glass cover at the distal end. The optical path is built up as a stack of glass lenses with an additional separated light channel integrated into the outer stainless steel pipe and main body. A connector for a light cable, usually with a thread that’s shape is dependent on the manufacturer, is placed on the side of the main body of the endoscope. Rigid endoscopes come with a variety of angles of view. The most common variants are 0° straight view and 30°, 45°, 70°, 90° side view. Some rigid endoscopes can have steerable fields of view or integrated zooming options.

For imaging, the camera can be connected at the proximal end of the standardized eyepiece. This allows the use of separated cameras with higher image quality and a significant focus and zoom range. Another advantage of a separate camera is the avoidance of a hot and wet sterilization process that is crucial for the camera’s electronics. Some manufacturers have autoclavable camera heads in their portfolio others still use wipe disinfection and/or sterile covers. The rigid endoscopes themselves can be sterilized with a standard hot steam procedure in the autoclave and have a long lifetime. Only kinking or hard beats on the shaft can destroy the sensitive glass lenses, especially for small diameter scopes. Most manufacturers include flexible dampers that hold the lenses to reduce this risk. The imaging quality is highly dependent on the whole imaging chain, including optics, camera, video processor, monitor, light source and light cable or enhancement technologies.

Additionally, endoscopes are frequently used with other equipment like trocars, guides, sheets or holders. To visualize compatibility to the user, colored rings can be added to the endoscope and the additional equipment. A wide range of particular types of rigid endoscopes is available for various use cases. For example, there are endoscopes with a steerable field of view, realized by a rotatable prism inside the tip or a short flexible part (Figure 2a). Zooming and focusing options can be integrated, and even high magnification to observe the tiniest structures, e.g., in contact endoscopy. Work channels for instruments, flushing, or even a lens cleaning mechanism can be integrated for some applications. For robotic applications, the cameras can be integrated directly into the tip of the endoscope. This allows a side-by-side placement of two cameras to achieve stereo view and realize depth perception.

Flexible endoscopes are used to examine anatomies that are hardly or not reachable with rigid endoscopes. Mainly they are entered via the natural human orifices following the shape of the anatomy. There are two types of these endoscopes, fiber endoscopes and video endoscopes. In fiber endoscopes, the optical channel is created by a bundle of glass fibers combined with a lens on the tip, defining the focus point and the field of view. The ocular with a standardized eyepiece is mounted on the proximal end. A light source can be connected to a light post on the main body to illuminate the field of view. Some additional fibers transmit the light in the bundle to the tip of the endoscope. Depending on the diameter and construction, the number of fibers is limited. Since every fiber represents one pixel, the image’s resolution is limited to the number of imaging fibers. This reduces the image quality. The significant advantage of fiber endoscopes is their thickness and flexibility. Endoscopes with 5000 fibers come with a diameter of only 0.5 mm and can tolerate a bending radius of less than 7 mm. Endoscopes with 10,000 fibers and a diameter of 1 mm still reach a 10 mm bending radius [36]. Fiber endoscopes are mainly used to image very small anatomies, for example, in bronchoscopy or the upper urinary tract.

Video endoscopes have an integrated camera sensor on the tip. The image is transmitted via cables to the video processor of the imaging system. Diameters in everyday clinical use start from 2.9 mm for trans nasal applications up to 15 mm in Gastroenterology. The field of view illumination is usually realized by integrated light fibers connected to an external light source. The camera sensor provides at least HD image resolution and is combined with a lens stack for zoom and focus. Some video endoscopes can even realize magnifications up to 500 times with integrated optics.

The general design of flexible video endoscopes shows tubular steel braided or coiled support structure that is compression resistant but flexible for bending. The frontal part is usually more flexible and can have an integrated steering mechanism usually realized by two or more steering wires. Cables, light transmission fibers, steering wires and for some applications working, flushing, air or suction channels run inside the support structure. A rubber-like outer tube seals the inner part from the environment. The endoscope is operated with a multifunctional handgrip with wheels and knobs for steering, zoom, focus, image acquisition and light control. If a working channel is integrated, the access is usually placed on the distal part of the handgrip. The connection plug can integrate channels for flushing, air and suction. Sterilization of flexible video endoscopes is challenging and includes precleaning, manual or automatic chemical cleaning, drying and leakage testing. The sensitive rubber coat can be damaged during use and underlies aging over time, limiting the lifetime of flexible video endoscopes. Figure 3a shows different types of flexible endoscopes, and Figure 3b shows a detailed view of a connection plug, handgrip and different types of distal tips of flexible endoscopes.

Particular types of endoscopes are small imaging capsules (Figure 4b down left). These systems look like a pill with a diameter of approximately 11 mm and a length between 25 and 35 mm. The capsule can be swallowed and follows the natural path through the intestinal tract acquiring images. The capsule contains one or two cameras and LED lights and is powered by a stack of batteries providing energy for 8–12 h. The images are transmitted wireless to an antenna and recording system outside the body placed in a belt. The stored image series with up to 50,000 single shots can be evaluated after the examination with the help of particular software.

#### 2.5.2. Imaging Systems

The core of the imaging system is the camera sensor. Nowadays, the sensors of imaging systems used for endoscopic diagnostic and treatment observation provide at least High Definition (HD) resolution. HD refers to 1280 × 720, or 1366 × 768 pixels. FullHD is twice as large, at 1920 × 1080 pixels. Some companies have started to offer Ultra HD (UHD or 4 K) with a pixel number of 3840 × 2160, four times more pixels than FullHD. Both sensor types, CCD and CMOS, are on the market, but CMOS seems to be the technology of future generations. Some cameras combine two or more sensor arrays to improve sharpness and performance. However, image quality is not only linked to camera sensors. The whole imaging chain has to be optimally concerted. This includes the lenses, endoscope optics, and the monitor and important as well; a dirty or scratched endoscope cannot provide clear and sharp images. Additionally, the image documentation mode, video or static image, plays an important role and can influence the diagnostic value. Videos recorded at frame rates of 30 Hz can contain a larger variety of blurring, reflection or defocus.

For some applications, stereo cameras are beneficial for allowing 3D visualization. This is mainly used in laparoscopic and robotic surgeries where haptic feedback and depth perception is limited. Some examples of external camera heads and integrated camera solutions are shown in Figure 4a,b.

The acquired camera sensor data is transferred to a video processing unit for conversion into an image. The video processor optimizes the image or real-time video depending on the selected presetting’s like white balance, color display mode, reflection reduction or image rotation and transfers it to a screen. Additionally, scenes and images can be stored for later diagnostics or documentation. Some video processors provide, in combination with distinct light sources, enhancement technologies like narrow band imaging (NBI), autofluorescence (AF) or flexible spectral imaging color enhancement (FICE). Using a selected wavelength for illumination and filtering the acquired image can improve the visibility of fine structures, details, or distinct lesions [29,37,38,39,40,41,42]. Linked color imaging (LCI) and blue laser imaging (BLI) are image enhancement technologies using narrow band light. Blue and green color information are processed separately from the red channel and allow enhancement structures of interest [43].

Besides the video processor, another essential device in the endoscopic imaging system is the light source. Expressive images can only be acquired with sufficient illumination. The light color, intensity, and homogeneous distribution are relevant parameters for light sources. In the past, Halogen or Xenon-based light sources were familiar but were expensive and with a limited lifetime. New systems today usually come with LED-based light sources with long lifetimes. They offer high power light output and are energy-saving compared to the older technologies. The light source can be coupled with the video processing unit. This allows automatic adaption of light intensity based on the resulting image to avoid overexposure and reflection. Also, the light source must interact with the video processor for enhancement techniques. Light sources for these technologies offer illumination in different selectable wavelengths by using colored LED (BLI,) or, in the case of Xenon and Halogen lights, by adding filters (NBI). Combinations of both Technologies are also available on the market (LCI, linked color imaging).

#### 2.5.3. Additional Equipment

Endoscopic systems are used with several additional pieces of equipment depending on the application. For some, like urology, introducer sheets are helpful when systems have to be changed frequently. CO_2_ insufflation can create space for imaging and advancement of devices, for example, in laparoscopy and colorectal applications. Suction is helpful when ablation techniques are applied for smoke reduction; flushing and suction can clean the field of view or the surgery area. Multiple tools like graspers, forceps, slings, clips or injection devices are available for rigid and flexible endo therapeutic approaches. For ablation of lesions, high-frequency generators and probes or laser-based methods can be used. Some endoscopes are equipped or can be used with small ultrasound probes for multimodal imaging. For some applications, balloons for blockage, stabilization or balloon and spiral-based motors for more straightforward advancement are available. These multiple options made endoscopic diagnostic and treatment successful in this wide variety of applications.

### 2.6. Upcoming Trends

The level of integrated technology into endoscopic imaging systems is already high. Besides the cutting-edge systems, nearly all providers offer less costly workhorses for daily routine or in-office applications. Nevertheless, there is a clear trend to higher pixel numbers in imaging, similar to the non-medical market. Some providers already offer 4K imaging and visualization in their high-end systems. On the other hand, the trend of minimalization and integration will go on, leading to more multifunctional endoscopes with improved performance. This covers a larger focus range, zoom factors and light options or ends up in a cost reduction or smaller diameters for these devices. 3D imaging has not made its way to broad standard applications and is reserved for distinct procedures like robotic surgeries.

Another trend already seen in first applications is advanced postprocessing of the images. This can be used to correct motion, blurring or reduce image interferences caused by smoke in the field of view or pixel defects.

A clear trend is using artificial intelligence (AI) platforms to support endoscopy-based diagnostics. AI describes the ability of computer algorithms to learn and solve problems. In medical applications, AI uses algorithms for the detection and classification of features trained on annotated medical data. In the case of endoscopy, these data are images depicting healthy or diseased tissue. The more data is fed into the algorithm, and the better is the ground truth of the annotation, the more accurate is the outcome of the algorithm. AI is used today mainly in gastrointestinal endoscopy, improving cancer detection, diagnosis of infections, and detection of bleedings or polyps [44]. Figure 5 shows the AI platform of Fujifilm for colorectal applications. The main reason for the advanced use of AI in gastrointestinal endoscopy is the availability of large databases of images in gastrointestinal endoscopy. However, there are several catalogs of images in other medical fields available providing descriptions of the characteristic patterns of diseases [45,46,47]. Additionally, new annotated image repositories contribute to the big data needed for AI [48,49,50]. These data are an ideal starting point to train algorithms of artificial intelligence and include and preserve the gained experience of the user community. This technology can improve detection rates, assist physicians with less experience and help to automatize image analysis to reduce workload and speed up the clinical workflow [51]. Besides learning those tasks that physicians can already do well, AI is supposed to learn those where physicians have had only limited success [52] so far. Another upcoming field for AI is Surgical Data Science which aims to improve the quality of interventional healthcare and its value through the capture, organization, analysis and model of data [53]. Research is quite active and many companies are entering in this field today. Researchers and companies are developing substantial endoscopic video databases for using in two main fields: training + simulation, and context awareness in surgery [54].

Besides AI, the digitalization and storage of diagnostic data like endoscopic images allow objective evaluation of changes over time. For example, lesions can be described and classified in a first diagnosis, and disease progression can be detected and calculated automatically in later examinations. In the near future, more and more health records will be collected over patients’ lifetimes and merged into a patient individual database and model. Even though this option counts for all digitalized patient records, endoscopy, as one of the most used imaging modalities, is pioneering this trail to the “digital patient”, including prediction and prevention [55].

For some applications, external navigation is beneficial, for example, in orthopedics, neurosurgery or ENT procedures [56]. This opens up the option for registration to prior acquired datasets (CT, MRI), image overlay or virtual endoscopy to improve the identification of targets and risk structures [57,58]. However, due to the effort in setting up navigation before the procedure, the application is limited and did not find its way into a routine clinical application.

For mobile use, in-office use or in combination with tiny endoscopes, wireless camera systems and light sources are available [59]. The missing cables provide more freedom to operate. Nevertheless, image resolution and performance are not comparable to wire-driven systems.

Nowadays, single-use endoscopes [9,60,61] have entered the market. A single-use can reduce the risk of infections or cross-contamination. Another advantage is avoiding the time-consuming and costly sterilization procedure, maintenance, and repairs needed for multi-use systems. Single-use endoscopes are significantly cheaper than traditional scopes. The camera and LED-based light source are integrated into the tip. The systems can be connected to a multi-use monitor. Depending on the application and size, HD quality is available. The manufacturers promise similar performance like multi-use scopes.

One of the promising imaging technologies on the horizon is hyperspectral imaging. For endoscopic diagnostics, it is a technology still under research [62]. However, it allows the recording and analysis of spatial and spectral information. In addition, the light-tissue interaction such as absorption, scattering or fluorescence contains information about morphological and biochemical processes and can help to improve diagnostic accuracy [63].

Many research works are published on advanced endomicroscopy such as confocal microscopy, multiphoton microscopy or optical coherence tomography. These methods use laser light to scan superficial tissue layers at a depth of 0–3 mm on a microscopic level [64] and show great potential for noninvasive early cancer detection. Therefore, laser light with a small focus spot follows a scanning pattern sweeping over the tissue of interest. For OCT the scattering of the tissue in comparison to a reference signal is measured [65,66]. Confocal imaging depicts reflections or fluorescence of the tissue in the focused depth [67]. Multiphoton imaging uses short, intense laser pulses to create fluorescence of the anatomic structures [68]. Systems for advanced endomicroscopy are expensive and are not seen in the clinical routine for endoscopic applications now.

## 3. Summary

The research focused on technology analyzing the catalogs and websites of endoscopic system providers is challenging. The medical fields of application usually separate the offers, and detailed information on the technology inside is hard to get. For this reason, a combined data collection was performed, including the analysis of the offer of the endoscopic systems providers, discussions with clinical users, an internet and market report research and a publication screening. Thus, a collection of the state of the art of nowadays endoscopic imaging technology could be presented.

The future of endoscopic imaging is digital and interconnected. The utilized electronics will improve in size and performance, and new technologies will be implemented. Endoscopic imaging aims to acquire meaningful data, store and process this data for better presentation, and collect and connect data for automatized and comprehensive context evaluation. The permanent improvement of technical equipment in hospitals and practices combined with assistance systems will improve early detection and reduce the need for more extensive interventions. With the vision of the “digital patient” model in mind, endoscopy can provide digital and evaluable data for diagnosis and therapy, but also for monitoring. Therefore, technologies to document the procedures and for mapping the diagnostics on the digital patient model would be of interest to apply the full power of the upcoming artificial intelligence in diagnostics and prevention.

## Figures and Tables

**Figure 1 diagnostics-12-01262-f001:**
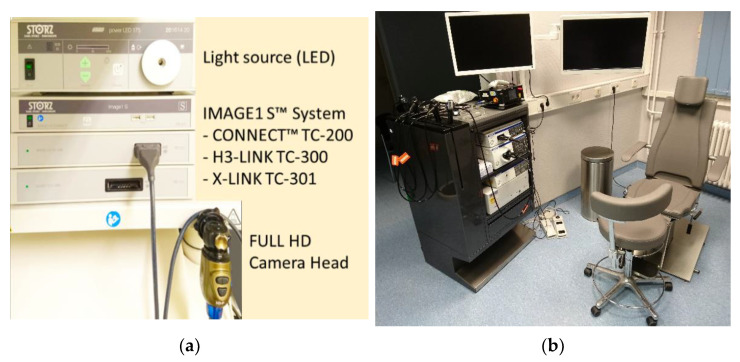
(**a**) Example of an endoscopic imaging system of KARL STORZ. The modular video processor IMAGE 1 allows the connection of various endoscopic devices. (**b**) Endoscopy workplace for ENT with OLYMPUS video processor, light sources and flexible endoscopes on the right.

**Figure 2 diagnostics-12-01262-f002:**
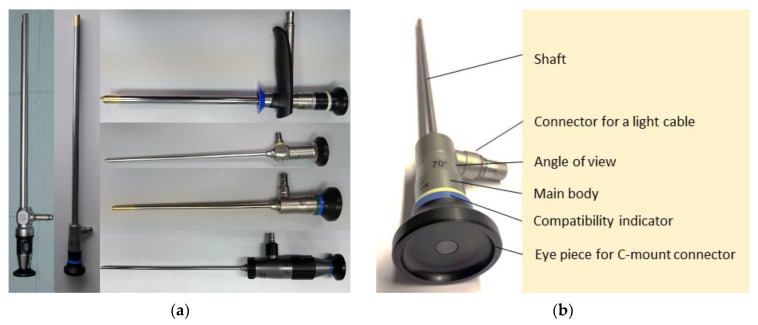
(**a**) Variants of rigid endoscopes, left: KARL STORZ EndoCAMeleon with the steerable field of view 0–120°, OLYMPUS 10 mm laparoscope 30 cm, right top-down: OLYMPUS 10 mm rigid laryngoscope with a detachable handle and 70° view, KARL STORZ 4 mm sinuscope 70° view, OLYMPUS 4 mm sinuscope 0° straight view, OLYMPUS 4 mm steerable multi view sinuscope (**b**) general components of a rigid OLYMPUS 4 mm sinuscope with 70° angled view.

**Figure 3 diagnostics-12-01262-f003:**
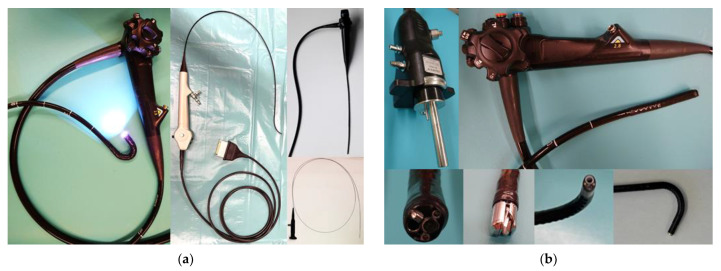
(**a**) Variants of flexible endoscopes: Fujifilm gastroscope, KARL STORZ cystoscope, OLYMPUS rhinolaryngoscope, flexible fiber endoscope (**b**) Flexible video endoscope components, upper row: Fujifilm connector plug to the imaging system, Fujifilm gastroscope handle, lower row: Fujifilm gastroscope tip, Fujifilm duodenoscope tip, KARL STORZ cystoscope tip, OLYMPUS rhinolaryngoscope tip.

**Figure 4 diagnostics-12-01262-f004:**
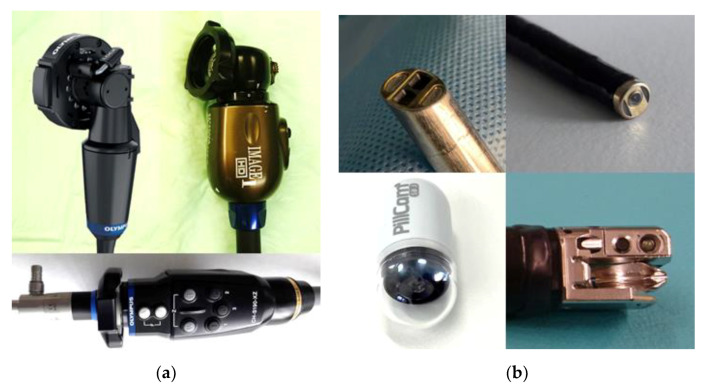
(**a**) Camera heads with C-mount connector, upper row: OLYMPUS pendulum camera, KARL STORZ H3P CCD camera, OLYMPUS CCD autoclavable camera (**b**) integrated cameras, upper row: Intuitive Da Vinci stereo camera, OLYMPUS rhinolaryngoscope camera, lower row: Medtronic PillCam with LED lights, Fujifilm duodenoscope camera.

**Figure 5 diagnostics-12-01262-f005:**
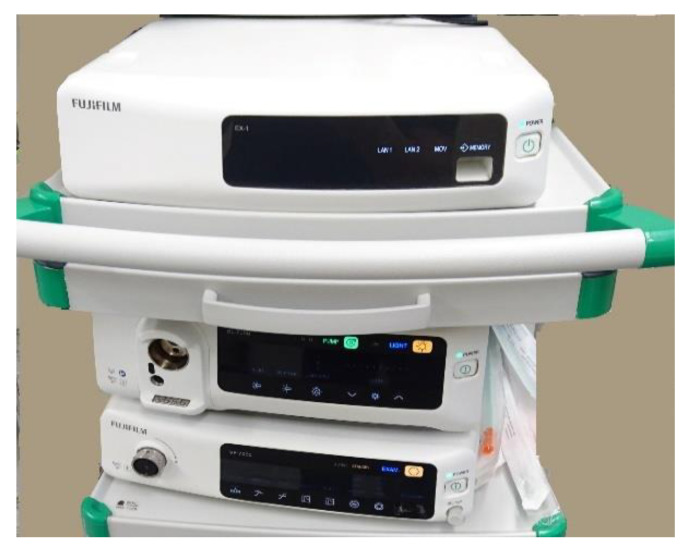
Fujifilm endoscopy system with EX1 CADEYE AI platform on top.

**Table 1 diagnostics-12-01262-t001:** Endoscopy systems providers (alphabetic order) and their fields of application.

Company	Fields and Portfolio	Key Technology, Highlights	Web
Ambu [9]	Pulmonology, ENT, gastroenterology, urology	single-use endoscopes full-HD	https://www.ambu.com (Accessed on 19 January 2022)
Arthrex, Inc., 1370 Creekside Blvd., Naples, FL 34108, USA [25]	Arthroscopy, endoscopy, open surgery, laparoscopy, endoscopes for ENT, rigid endoscopes	SYNERGY 4K Fully integrated surgery room SynergyID Near-Infrared Fluorescence 3-in-1 console including light documentation, video-processor NanoScope™ 3-in-1, chip-on-tip, single-use camera system	https://synergy.arthrex.com (Accessed on 22 December 2021)
ATMOS Medizin Technik GmbH & Co. KG [7]	ENT	CMOS Camera with detachable cable Stroboscope application	https://atmosmed.com (Accessed on 31 January 2022)
Auris Health [19]	Robotic-assisted bronchoscopy	MONARCH^®^ Platform for Robotic-Assisted Bronchoscopy	https://www.aurishealth.com/monarch-platform (Accessed on 21 December 2021)
B. Braun Melsungen AG, Aesculap [17]	Rigid endoscopes for laparoscopy, endoscopic gynecology, thoracoscopy, endoscopic vascular surgery in the pelvic region, neuro-endoscopy and arthroscopy	AESCULAP EinsteinVision 3.0 with 3D image quality with native Full HD resolution, impressive depth of field and high image contrast. Anti-fogging function Smoke reduction Red enhancement	https://www.bbraun.dehttp://www.endoscopy-catalog.com/en/index.html (Accessed on 22 December 2021)
Boston Scientific Corporation [16]	Flexible endoscope for gastrointestinal, endoscopic retrograde cholangiopancreatography (ERCP) Pulmonary endoscopes	The SpyGlass™ DS Direct Visualization System, a single-use, single-operator digital cholangioscope, enables physicians to see color images from inside the biliary, hepatic and pancreatic ducts and also interventions like electrohydraulic lithotripsy (EHL) or argon plasma coagulation (APC)	https://www.bostonscientific.com/en-US/products/single-use-scopes/spyglass-ds-direct-visualization-system.html (Accessed on 22 December 2021)
CMR surgical [15]	Robotic surgery	Versius surgical robot including integrated stable 3D HD visualization	https://cmrsurgical.com (Accessed on 22 December 2021)
CONMED Corporation [14]	Rigid endoscopes for arthroscopy and laparoscopy	All-In-One Multi-Specialty 4K System provides extremely bright images with incredible definition and virtually no pixilation. Efficient LED Light Bulbs Autoclavable Camera Heads Wi-Fi Connection to Hospital EMR Networks	https://www.conmed.com/en (Accessed on 22 December 2021)
Depuy Synthes [6]	Arthroscopy, endoscopy	PUREVUE™ Visualization System, HD and 4K displays, cameras, endoscopes	https://www.jnjmedicaldevices.com/en-US/companies/depuy-synthes (Accessed on 31 January 2022)
Fujifilm Holdings Corporation [21]	Flexible endoscopes for gastrointestinal, biliary intervention, bronchoscopy, Endoscopic ultrasound	7000er series: Endoscope system with diagnosis and usability contains the light source which features high-intensity 4 LED lights. CAD Eye AI for detection and characterization, double-balloon endoscope diagnosis for diseases of the small intestine.	https://www.fujifilm.com/products/medical/endoscopy (Accessed on 16 October 2021)
Intuitive Surgical [20]	Robotic surgical systems, including endoscopy	highly-magnified 3DHD Imaging, lightweight focus-free endoscope, advanced digital optics in tip-mounted cameras, Firefly near-infrared fluorescence imaging	https://www.intuitive.com (Accessed on 21 December 2021)
KARL STORZ GmbH & Co. KG [22]	Rigid and flexible endoscope, neurosurgery, oral and maxillofacial surgery, otorhinolaryngology (ENT), anesthesiology and emergency medicine cardiovascular surgery, thorax, mediastinum, plastic surgery, gastroenterology, laparoscopy, gynecology, urology, proctology, arthroscopy spine surgery, microscopy, pediatrics	IMAGE1 S™ 4U modular platform 4K technology, IMAGE1 S™ 3D	https://www.karlstorz.com/de/en/human-medicine.htm (Accessed on 16 March 2021)
Medtronic [23]	Capsule endoscopy, rigid scopes for laparoscopy, robotic surgery systems	PillCam, GI Genius™ intelligent endoscopy module (AI Gastro), EleVision™ IR Platform: Can be used for both open and laparoscopic procedures Hugo robotic surgery system	https://www.medtronic.com/covidien/en-us/products/capsule-endoscopy.html (Accessed on 16 March 2021)
Mindray [13]	Rigid scopes for laparoscopy	HyPixel TM U1 4K Endoscope Camera System Wide color gamut, 3–200 mm depth of field	https://www.mindray.com/en/productlist/LaparoscopicProducts.html (Accessed on 11 January 2022)
Olympus [24]	Gastroenterology, ENT, general surgery, gynecology, neurosurgery, pulmonology, urology, capsule endoscopy	EVIS X1 4K advanced endoscopy system, TXI new white light—Explore texture and color enhancement imaging, ENDO-AID CADe to embrace computer-aided detection, RDI safeguard for endoscopic therapy—Experience red dichromatic imaging, EDOF full focus with extended depth of field, Brighter NBI, Dual-focus mechanism, ENDOCAPSULE EC-10 System	https://www.olympus-europa.com/medical/en/Home (Accessed on 15 March 2021)
Ottomed Endoscopy [18] and Mitra Medical Services	Flexible scopes and imaging systems for gastroenterology, pulmonology, urology, laparoscopy	SmartEye-II Plus, HD+ platform with super bright LED-At-Tip light source, Tactile Insertion Tube (TIT), Touch screen interface and extreme Close-Up Endoscopy (eCUE) technology	https://www.ottomed.com (Accessed on 15 March 2021)
PENTAX Medical [12]	Flexible scopes for gastrointestinal, ultrasound gastroscope, bronchoscopy ENT video endoscopes + fiberscopes, urology cystoscopes+ ureteroscopes	I-scan image processing for digital image enhanced endoscopy (IEE), Real-time virtual chromoendoscopy, Advanced HD endoscopes, DISCOVERY™ artificial intelligence for support detection of unremarkable lesions, Confocal laser endoscopy (CLE)	https://www.pentaxmedical.com/pentax (Accessed on 11 January 2022)
Richard Wolf Medical Instruments [11]	Rigid and flexible endoscopes for urology, general surgery, gynecology, orthopedics, pulmonology, thoracic surgery, mediastinum, spine surgery, proctology, pediatrics shockwaves, ENT, Single use endoscopes	ENDOCAM Logic 4K camera controller, Logic 4K camera head real-time ICG/NIR fluorescence imaging, RIWO D-URS single use endoscopes Spreadable mediastinoscope	https://www.richard-wolf.com/en (Accessed on 11 January 2022)
Robotics Surgical [8]	Flexible robotic system for visualization and surgical site access through the oral or rectal access path	Flex^®^Robotic system containing HD flexible endoscope with steerable outer skeleton for guidance and channels for instruments	https://robotics-surgical.com/flex-roboticshttps://medrobotics.com (Accessed on 31 January 2022)
Stryker [10]	Imaging system for all minimally invasive procedures	1588 AIM (Advanced Imaging Modalities) platform, DRE Dynamic Range Enhancement High-definition camera system, Clarity: Real-time video enhancement 4K 32′ surgical display	https://www.stryker.com/us/en/endoscopy.html (Accessed on 11 January 2022)

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
