# Peer review of "Endoscopic Imaging Technology Today"

_diagnostics, 2022, doi:10.3390/diagnostics12051262_

Round 1

Reviewer 1 Report

I read this review with interest. It was well-written with overview. However, description of AI technique would be insufficient. 

Author Response

Thank you for your interest on the paper. Following the review comments, some modifications have been performed to the revised version of the manuscript.

First of all, we have gone through additional English language editing and hope that it now qualifies for good readability. Second, we have added more information on the description of the AI technique in Section 2.6 (lines 397-495).

Reviewer 2 Report

It is hard for me to see what this paper adds to the current scientific  literature. The fact that 12 totally different fields of applications are presented together makes the analysis of each one rather superficial and not supported by sufficient clinical evidence.

Most of the references in the bibliography section come directly from the companies websites or are just links to websites like "https://www.verywellhealth.com/". The data about clinical applications of endoscopy is very scarce.

However, the concept of "digital patient" stressed in the discussion section seems interesting and could be worth more space in the article.

Author Response

Thank you for the valuable review comments on the paper.

The intention of the paper is to give an overview of modern endoscopic technology's state of the art systems and the providers on the market. Thus, the focus is more on the technical design and features of instruments and endoscopic systems that change with the field of application. The focus is not on the detailed description of the medical procedures. The target groups of this paper are readers from the medical field as well as from medical engineering who are interested in technology and want to be informed what other medical disciplines do. Up to our knowledge, such an overview of endoscopy systems and the technological features does not exist in the current literature.

Following the review comments, some modifications have been performed to the revised version of the manuscript. First of all, we have gone through additional English language editing and hope that it now qualifies for good readability. Second, we have changed references 26 and 27 to definite books in the field of endoscopy. We kept the Web references of the endoscopic system providers since these are the sources where the technology portfolio is described. Third, we added some more details on the digital patient concept (lines 428-432). This concept is about the collection and evaluation of patients' health records. Imaging like endoscopy is one of the significant sources that can contribute to this individual database. For this paper, it is only an option shown in the future trends but is not the main focus of the overview. Finally, we added a reference (53) where more information about the digital patient concept is available.

Reviewer 3 Report

The review on modern endoscopic instrument has been prepared well. The literature search is sufficient and supports the main content. The figures also look professional with adequate caption. Endoscopic confocal, multiphoton, and OCT may be worth mentioning. Generally speaking, the review article's technical writing is impressive and will be useful to the field. "Accept" is recommended.

Author Response

Thank you for your interest on the paper. Following the review comments, some modifications have been performed to the revised version of the manuscript.

First of all, we have gone through additional English language editing and hope that it now qualifies for good readability. You are correct in asking for advanced endomicroscopy technologies as a future trend. We have added more information on confocal microscopy, multiphoton microscopy and optical coherence tomography in section 2.6 (lines 456-465).

Reviewer 4 Report

The authors cover current and past state-of-the-art endoscopic techniques available worldwide in this interesting review. The analysis is well organized and informative, I have only a few comments I believe could improve paper's quality. 

Minors:
1. Article needs a considerable revision in used English language.
2. The number of self-citations is beyond the standard. Despite the great experience of the first author, I believe there are other valuable experts we could include to improve this manuscript.

Author Response

Thank you for your interest on the paper. Following the review comments, some modifications have been performed to the revised version of the manuscript.

First of all, we have gone through additional English language editing and hope that it now qualifies for good readability. Second, we have extended some paragraphs in section 2.6 and added more independent references of technical experts in the field.

Round 2

Reviewer 1 Report

I have no comment.